# Genomic SELEX Reveals Pervasive Role of the Flagella Master Regulator FlhDC in Carbon Metabolism

**DOI:** 10.3390/ijms24043696

**Published:** 2023-02-12

**Authors:** Hiraku Takada, Kaede Kijima, Akira Ishiguro, Akira Ishihama, Tomohiro Shimada

**Affiliations:** 1Micro-Nano Technology Research Center, Hosei University, Koganei, Tokyo 184-0003, Japan; 2Faculty of Life Sciences, Kyoto Sangyo University and Institute for Protein Dynamics, Kamigamo, Motoyama, Kita-ku, Kyoto 603-8555, Japan; 3School of Agriculture, Meiji University, Kawasaki, Kanagawa 214-8571, Japan

**Keywords:** transcription factor, flagella master regulator FlhDC, genomic SELEX (systematic evolution of ligands by exponential enrichment) (gSELEX), carbon metabolism, *Escherichia coli*

## Abstract

Flagella are vital bacterial organs that allow microorganisms to move to favorable environments. However, their construction and operation consume a large amount of energy. The master regulator FlhDC mediates all flagellum-forming genes in *E. coli* through a transcriptional regulatory cascade, the details of which remain elusive. In this study, we attempted to uncover a direct set of target genes in vitro using gSELEX-chip screening to re-examine the role of FlhDC in the entire *E. coli* genome regulatory network. We identified novel target genes involved in the sugar utilization phosphotransferase system, sugar catabolic pathway of glycolysis, and other carbon source metabolic pathways in addition to the known flagella formation target genes. Examining FlhDC transcriptional regulation in vitro and in vivo and its effects on sugar consumption and cell growth suggested that FlhDC activates these new targets. Based on these results, we proposed that the flagella master transcriptional regulator FlhDC acts in the activation of a set of flagella-forming genes, sugar utilization, and carbon source catabolic pathways to provide coordinated regulation between flagella formation, operation and energy production.

## 1. Introduction

Flagella motility helps bacteria reach favorable environments and is vital in substrate adhesion, biofilm formation, and virulence processes [1]. Bacterial flagellum synthesis genes form an ordered cascade in which the expression of one gene at a given level requires the transcription of another gene at a higher level [2,3]. This regulatory cascade includes three gene classes. In *Escherichia coli*, a model bacterium, class I genes form the *flhDC* master operon at the top of the hierarchy, encoding an FlhDC (more specifically, FlhD_4_C_2_) transcriptional activator of class II gene expression [4]. Most class II genes encode flagella export system and basal body components. The *fliA* gene at this second level encodes a sigma factor, σ^28^ (or RpoF), specific for flagella genes [5,6]. σ^28^ and the anti-sigma factor FlgM positively and negatively regulate class III operons, respectively [7]. The cell retains the anti-sigma factor until the flagella basal body and hook are completed [8].

The *flhDC* operon is at the top of the flagella regulatory cascade and encodes the FlhDC activator required for the expression of all other genes in the flagella regulon. FlhD and FlhC form a heterohexamer (D_4_C_2_) that binds upstream of class II promoters and stimulates class II gene transcription [9]. The FlhD and FlhC proteins are class I transcription factors (TFs). The C-terminal domain of the RNA polymerase α-subunit is essential for transcriptional activation [10]. The DNA fragment bound by FlhDC, called the FlhDC-box, contains two 16 bp inverted repeats, with a 10–12 bp spacer between them [11,12,13,14]. The *flhDC* gene is under the control of multiple TFs that integrate various environmental cues (listed in the RegulonDB database (regulondb.ccg.unam.mx). Additionally, see the review in Ishihama et al., 2016 [15]). CRP (TF for secondary carbon source catabolism), H-NS (for nucleoid-associated multifunction), and QseB (for quorum-sensing) are known activators. AcrR (for acriflavine resistance), Fur (for iron homeostasis), FliZ (for curli and motility), HdfR (for flagella master operon *flhDC*), IHF (for nucleoid-associated multifunction), LrhA (for fimbriae synthesis), MatA (for flagellum biosynthesis), OmpR (for outer membrane proteins), RcsAB (for capsule synthesis), and YjjQ (for flagella synthesis) are known *flhDC* repressors.

Researchers have attempted to identify the FlhDC regulon using ChIP-seq to identify FlhDC-binding sites across the *E. coli* genome [14] and microarray analysis to determine the effect of *flhDC* deletion and overexpression [12,16] on global gene expression in vivo. These genome-wide approaches have revealed novel FlhDC targets. However, it is difficult to distinguish between the direct and indirect effects of FlhDC. For instance, the direct targets of TFs generally represent only minor fractions of the genes detected by microarray and RNA-seq analyses. Especially for FlhDC, genes encoding other transcription regulators are organized together to form a complex regulatory network downstream of a specific regulator [17,18]. In vivo analyses can also be difficult because a set of regulatory proteins involved in single promoter regulation often compete for binding at overlapping DNA sites [15]. 

In this study, we performed genomic SELEX (gSELEX) to screen FlhDC-recognized genomic DNA sequences to avoid the problems associated with in vivo experiments describe above. The original SELEX screening method uses randomized synthetic oligonucleotides to identify the target DNA sequence of a tested protein. However, after an in silico computer-based homology search for consensus sequences, it is difficult to uncover the whole set of target genes from the entire genome because one must distinguish between true and false positives. On the other hand, the gSELEX screening system uses genome fragments with all FlhDC target DNA sequences. It was developed to directly identify DNA sequences recognized in vitro by DNA-binding TFs [19,20] and target promoters, genes, and operons under the direct control of each TF. We have successfully applied the gSELEX screening system to identify the regulatory targets of more than 150 TFs from a single species: *E. coli* K-12 W3350 [15,21]. Here, we used a gSELEX-chip system to identify FlhDC-binding sites in the *E. coli* K-12 genome. The results indicated that FlhDC regulates genes involved in flagella formation, sugar utilization, and carbon source metabolism.

## 2. Results

### 2.1. Identification of FlhDC Regulation Targets per gSELEX-Chip Screening

We performed gSELEX screening using an improved method to identify a set of FlhDC-binding sequences [19,20]. Purified His-tagged FlhDC was mixed with a mixture of 200–300 bp *E. coli* genomic fragments, and FlhDC-bound DNA fragments were affinity-isolated. The original substrate mixture of original genomic DNA fragments showed smear PAGE bands. However, after six cycles of gSELEX, DNA fragments with a high FlhDC affinity were enriched, forming sharper bands on the PAGE gels (Appendix A).

gSELEX fragments and the original DNA library were labeled with Cy3 and Cy5, respectively, to uncover a comprehensive set of FlhDC-controlled targets. A mixture of fluorescently labeled samples was hybridized onto an *E. coli* DNA tiling microarray [22]. The ratio of the fluorescence intensity between the FlhDC sample and the original DNA library of each probe was measured and plotted against the corresponding position along the *E. coli* K-12 genome. The extent of FlhDC binding correlated with its FlhDC affinity. After setting the cutoff level to twenty, 17 peaks were identified in the gSELEX-chip pattern (Figure 1), and 38 additional peaks were identified by decreasing the cutoff level to ten (Table 1). Of these 55 high-level binding peaks, 18 FlhDC-binding sites were located within spacers of bidirectional transcription units (Table 1, type-A), and 26 FlhDC-binding sites were located within spacers upstream of one ORF but downstream of another ORF (Table 1, type-B). Eleven FlhDC-binding sites were located within the genes or downstream of both transcription units (Table 1, type-C). Hence, we predicted that the total number of FlhDC regulatory targets was between at least 44 (18 type-A and 26 type-B) and 62 (18 × 2 type-A and 26 × 1 type-B).

Currently, the RegulonDB database lists 20 genes or operons as FlhDC targets ([23,24]. regulondb.ccg.unam.mx (accessed on 1 December 2022)). gSELEX screening detected seven of these targets above the cutoff level of ten (Table 1), with the highest peak intensity at 54 for the *fliE* gene, which encodes a component of the flagellum hook-basal body subunit. Notably, at this cutoff setting, many known FlhDC targets involved with the flagella genes, including the *flgAMN, flgBCDEFGHIJ, fliAZ-tcyJ*, and *fliLMNOPQR* operons, were detected (Table 1). However, the known target, the *fliDST* operon, which is critical in flagellin biosynthesis and assembly, showed a peak intensity of six.

### 2.2. Verification of FlhDC Consensus Recognition Sequence 

Using the DNA-binding sequence information from a set of FlhDC target sequences, a 42–44 bp palindromic sequence consisting of AAYGSSNNAAATAGCG-N_10-12_-CGCTATTTNNSSCRTT was proposed as the consensus recognition sequence for FlhDC [11,12,13,14]. As we obtained 49 new FlhDC-binding sites, the FlhDC consensus sequence was analyzed using the MEME program [25] to identify the FlhDC-box sequence in the newly targeted regions. We identified the same 42–44 bp sequence in 44 of the 55 FlhDC target regions (Table 1), and the high conservation of the A_1_, A_2_, G_4_, A_9_, A_10_, A_11_, T_12_, and A_13_ bases was also consistent (Figure 2) [12]. These results suggest that the novel target genes of FlhDC identified by the gSELEX screening was valid.

### 2.3. FlhDC Binding In Vitro to the New Target Genes Involved in Sugar Utilization and Carbon Source Metabolism

FlhDC (formerly FlbBl) was originally identified as a flagella operon regulator [26]. Subsequently, several research groups have performed comprehensive in vivo analyses, including transcriptome and ChIP-seq analyses [11,12,14]. FlhDC is the principal regulator of bacterial flagellum biogenesis and swarming migration, and a microarray showed that it regulates several non-flagella genes. However, there was no evidence that these genes were directly regulated by FlhDC. gSELEX screening identified FlhDC-binding sites in several promoter regions of genes involved in sugar utilization and carbon source metabolism (Appendix A), in addition to known FlhDC target genes involved in flagella formation (Appendix A). Genes involved in sugar utilization, such as the *ptsHI-crr* operon (phosphoenolpyruvate: sugar phosphotransferase system (PTS) for PTS sugar utilization), *fruBKA* operon (fructose-specific PTS system and fructose utilization), and *setB* (sugar transporter), and genes involved in carbon source metabolism, such as *pfkA* (6-phosphofructokinase (I), the *epd-pgk-fbaA* operon (D-erythrose-4-phosphate dehydrogenase, phosphoglycerate kinase, fructose-bisphosphate aldolase (II), *adhE* (acetaldehyde-CoA dehydrogenase/alcohol dehydrogenase), the *aceBAK* operon (malate synthase, isocitrate lyase, isocitrate dehydrogenase kinase/phosphatase), and *iclR* (isocitrate lyase TF), were included in the list of predicted FlhDC targets (Table 1, Appendix A). 

We focused on a set of genes involved in sugar utilization and carbon source metabolism to verify this novel role of FlhDC and analyzed the effects of FlhDC on gene regulation. We examined the in vitro binding of purified FlhDC complexes to sequences isolated by gSELEX screening using PAGE to confirm FlhDC–DNA complex formation. Upon increasing FlhDC supplementation, DNA fragments from the probes of known FlhDC targets detected in the gSELEX-chip formed FlhDC–DNA complexes in an FlhDC concentration-dependent manner (Figure 3a,b). The new targets were tested under the same conditions. FlhDC–DNA complex formation was observed for all tested probes depending on the FlhDC concentration (Figure 3c–i). In addition, FlhDC consensus sequences were identified in all the binding regions (Table 1). In contrast, the *purH/rrsE* intergenic region, the reference DNA added as a negative control (Appendix A), did not form FlhDC–DNA complexes under the same conditions (Figure 3j). These results indicated specific binding of all seven FlhDC target sequences.

### 2.4. In Vivo Transcription Regulation of the Set of FlhDC Target Genes Involved in Sugar Utilization and Carbon Source Metabolism

We performed a Northern blotting analysis to examine the possible influence of FlhDC on target promoters detected in vitro based on FlhDC-binding sites and determined mRNA levels in vivo for each of the predicted FlhDC target genes in the presence or absence of FlhDC. Wild-type *E. coli* K-12 and the *flhD*-deleted mutant grown in M9-glucose (0.2%) medium and total RNA was extracted in the exponential phase. The mRNA levels of individual FlhDC target genes were analyzed by Northern blotting analysis. The mRNA levels of *fliAZY* and *flgBCDEFGHIJ*, known targets for which FlhDC acts as an activator, were detected using *fliA* and *flgB* probes, respectively. Both transcripts decreased markedly in the *flhD* knockout mutant (Figure 4). *ptsHI-crr, fruBKA*, *setB, pfkA, epd-pgk-fbaA, aceBAK,* and *iclR* mRNA levels also significantly decreased in the *flhD* mutant (Figure 4), implying that FlhDC has an activating role in these transcription units. *adhE* mRNA levels slightly decreased, indicating that FlhDC also acts as an *adhE* activator. *metH*, located in the divergence from *iclR*, was not affected, suggesting that FlhDC binding between the *iclR/metH* intergenic region activates *iclR*. *pfkA* encodes 6-phosphofructokinase I, a key rate-limiting enzyme in the glycolysis pathway [27,28]. These results suggest that FlhDC directly activates genes involved in flagella formation and carbon source metabolism, mainly the PTS sugar uptake and glycolysis pathways.

### 2.5. The Effects of FlhDC on Sugar Utilization and Cell Growth

Transcriptional regulation analyses suggested that FlhDC activates a set of carbon source metabolism genes involved in glucose uptake (*ptsHI-crr*), fructose uptake and utilization (*fruBKA*), and glycolysis (*pfkA, pgk, fbaA*) (Figure 4). Cell growth and sugar consumption of wild-type and *flhD*-deficient strains in glucose and fructose sole carbon source media were observed to evaluate the effects of FlhDC on glucose and fructose uptake and catabolism. After 8 h of incubation in the glucose sole carbon source medium, the cell density of the wild-type strain reached OD_600_ = 0.85, and the glucose concentration decreased to 0.1% (Figure 5a). The OD_600_ was less than 0.3 in the *flhD-*deficient strain, and approximately 0.15% of the glucose remained. After 12 h of incubation, the wild-type strain entered the stationary phase, and the glucose in the medium was depleted, whereas the *flhD*-deficient strain continued to grow and left 0.03% of the glucose remaining. The effect of *flhD* deficiency on the fructose medium was similar to that on the glucose medium, with slower growth and more fructose remaining in the medium in the *flhD*-deficient strain than in the wild-type strain (Figure 5b). Note that there was no difference in cell growth between the wild-type and *flhD* deficient strains on LB medium (data not shown). These results suggest that *flhDC* stimulates energy production by accelerating the uptake and catabolism of PTS sugars, such as glucose and fructose.

## 3. Discussion

Motility is necessary for microorganisms to adapt to environmental changes, and the precise control of flagella formation is critical. Flagella formation in *E. coli* has been well-described [2,3]. The class I flagella master regulator FlhDC initiates the signal cascade mechanism for flagella formation. FlhDC-activated class II genes include those involved in flagella basal body formation and flagella sigma factors, RpoF (σ^28^). RpoF induction stimulates class III genes to assemble the hook and flagella filaments and complete flagella formation. Comprehensive in vivo analysis has been used to uncover the direct regulation of a set of genes involved in flagella formation [12,14]. Hitherto, microarray analysis showed that FlhDC regulates several non-flagella genes and/or operons involved in metabolism [11,16], such as the *glpABC* operon for glycerol degradation, the *gltI-sroC-gltJKL* operon (glutamate/aspartate transporter), *mdh* (malate dehydrogenase), the *mglBAC* operon (D-galactose transporter), the *napFDAGHBC-ccmABCDEFGH* operon (nitrate reductase), and the *nrfABCDEFG* operon (nitrite reductase) (listed in the RegulonDB database as FlhDC targets). However, the direct effect on these genes has not yet been determined, and only the *nrfABCDEFG* operon was detected in our gSELEX screening among the FlhDC low-intensity peaks (intensity level of eight; data not shown). In this study, we used a gSELEX analysis, which can identify direct target DNA sequences in vitro, to determine the genomic regulatory mechanisms of FlhDC. The results showed that previously unknown targets of class II flagellum-forming genes are involved in carbon source metabolism (Figure 1, Table 1), and FlhDC activates these genes (Figure 3 and Figure 4). As for the effects of these regulations on the phenotype, FlhDC stimulated the consumption of glucose and fructose and accelerated cell growth (Figure 5).

Novel genes regulated by FlhDC affect sugar utilization and metabolism of carbohydrate sources. The *ptsHI-crr* operon imports PTS sugars, such as glucose and fructose, and the *fruBKA* operon controls fructose import and utilization [29]. After uptake, sugars are catabolized in the glycolysis pathway and used for energy production such as ATP and NADH [28]. In the glycolysis pathway, the conversion of D-fructose 6-P to D-fructose 1,6-P2 via 6-phosphofructokinase is crucial for regulating the activity of the glycolysis pathway because this reaction is irreversible. *E. coli* has two 6-phosphofructokinase genes: *pfkA* encoding 6-phosphofructokinase 1 (PFK I), and *pfkB* encoding 6-phosphofructokinase 2 (PFK II). More than 90% of the phosphofructokinase activity in wild-type *E. coli* can be attributed to PFK I [27], suggesting that *pfkA* transcriptional activation via FlhDC has a marked effect on glycolysis activation. The *epd-pgk-fbaB* operon, activated by FlhDC, also contains *pgk-*encoded phosphoglycerate kinase and *fbaB-*encoded fructose-bisphosphate aldolase, enzymes for reversible reactions, suggesting that an increased carbon flux in the glycolysis pathway also occurs. Two pathways that metabolize the glycolysis product acetyl coenzyme A (acetyl-CoA) from sugars, the ethanol fermentation and glyoxylate pathways, were also identified as targets of FlhDC. *adhE* encodes fused acetaldehyde-CoA dehydrogenase and alcohol dehydrogenase. Thus, this enzyme catalyzes sequential two-step reactions, converting acetyl-CoA + NADH → acetaldehyde + coenzyme A + NAD^+^ and acetaldehyde + NADH → ethanol + NAD^+^. This multifunctional enzyme (AdhE) acts under anaerobic conditions to re-oxidize NADH to NAD^+^ because the glycolysis pathway requires NAD^+^ for sugar catabolism. FlhDC may activate *adhE* to maintain glycolysis activity under anaerobic conditions and energy production for flagella. In contrast, under aerobic conditions, the TCA cycle consumes an acetyl-CoA-derived carbon source to produce NADH for ATP production via an oxidative electron transfer system. The glyoxylate pathway is an anaplerotic reaction that enables *E. coli* to use acetyl-CoA as a carbon substrate. It consists of two enzymes, isocitrate lyase encoded by *aceA,* and malate synthase encoded by *aceB.* Isocitrate dehydrogenase kinase/phosphatase controls the activity of *icd-*encoded isocitrate dehydrogenase, a key enzyme for switching shifts between the TCA and glyoxylate pathways. FlhDC may activate the *aceABK* operon to maintain the activity of the TCA cycle under aerobic conditions to produce NADH for ATP production. On the other hand, this study focused on a group of genes in which FlhDC bound to the promoter region, and several binding regions were identified on the ORF (Table 1). It has been reported that transcription factors bind on ORFs [30] and examining these effects of these bindings may reveal new regulatory mechanisms.

Flagella synthesis and function are costly for cells [2,31]. Recently, it was estimated that all the flagella on a cell consume 10.2% of the intercellular ATP produced by *E. coli* (5% for construction and 5.2% for operating costs) [32]. Hence, we propose that cooperative transcription regulation of flagella and energy production (sugar utilization and carbon source metabolism) occur via the flagella master regulator FlhDC. Although this study examined the effects of FlhDC on a set of genes involved in central carbon metabolism with high FlhDC-binding strength using gSELEX screening, various other genes involved in carbon source metabolism and other biological functions were identified as novel FlhDC targets. Further studies could validate the effects of these genes to accelerate our understanding of the role of FlhDC in the entire *E. coli* genome.

## 4. Materials and Methods

### 4.1. Bacterial Strains and Plasmids

*E. coli* K-12 W3110 type-A [33] were used as the DNA source to construct the FlhD and FlhC expression plasmids. The *E. coli* K-12 W3110 type-A genome was also used to construct the DNA library required for gSELEX screening. *E. coli* DH5α cells were used for plasmid amplification, and *E. coli* BL21 (DE3) cells were used for FlhD and FlhC expression. *E. coli* BW25113 [34] and its *flhD* single-gene knockout mutant JW1881 [35] were obtained from the *E. coli* Stock Center (National Bio-Resource Center, Chiba, Japan). The pET21a(+) plasmid was used to construct the FlhD and FlhC co-expression plasmids. Cells were grown in an LB medium at 37 °C with constant shaking at 150 rpm. When necessary, 20 μg mL^−1^ of kanamycin or 50 μg mL^−1^ of ampicillin were added to the medium. Cell growth was monitored by measuring the turbidity at 600 nm.

### 4.2. Purifying the FlhDC Protein

A pFlhDC plasmid for expressing and purifying FlhDC was constructed according to the standard procedure, with slight modifications [19]. Briefly, FlhDC-coding sequences were amplified via PCR with a pair of primers (GTGGGACATATGCATCATCATCATCATCATCATACCTCCGAGTTGCTGAA and TACCGCTGCGCGGCCGCTCAGTGGTGGTGGTGGTGGTGAACAGCCTGTACTCTCTGTT) using *E. coli* K-12 W3110 genomic DNA as a template. They were inserted into the pET21a (+) vector (Novagen, Vadodara, India) between the NdeI and NotI sites, with the addition of a His x6 tag to the C-terminus of *flhC*, leading to the construction of the pFlhDC. The pFlhDC expression plasmid was transformed into *E. coli* BL21 (DE3) cells. Transformants were grown in an LB medium, and FlhD and FlhC were co-expressed by adding IPTG during the exponential phase. The FlhDC protein complex was purified by affinity purification using a NTA agarose column. The affinity-purified FlhDC protein complex was frozen in the storage buffer at −80 °C until use. Protein purity was greater than 90%, as determined by SDS-PAGE.

### 4.3. gSELEX Screening of FlhDC-Binding Sequences

gSELEX screening was performed as previously described [19,20]. Briefly, a mixture of DNA fragments from the *E. coli* K-12 W3110 genome was prepared by sonicating purified genomic DNA and cloned into a multi-copy plasmid, pBR322. For each gSELEX screening, the DNA mixture was regenerated by PCR. For gSELEX screening, 5 pmol of the DNA fragment mixture and 200 pmol FlhDC were mixed in a binding buffer (10 mM Tris-HCl, pH 7.8 at 4 °C, 3 mM magnesium acetate, 150 mM NaCl, and 1.25 mg/mL bovine serum albumin). The SELEX cycle was repeated six times to enrich the FlhDC-binding sequences. SELEX fragments were mapped along the *E. coli* genome using a gSELEX-chip system with a 43,450-feature DNA microarray [30]. The gSELEX sample obtained using FlhDC and the original genomic DNA library were labeled with Cy3 and Cy5, respectively. Following sample hybridization to the DNA tilling array (Agilent Technologies, Santa Clara, CA, USA), the Cy3/Cy5 ratio was measured, and the peaks of the scanned patterns were plotted against the positions of the DNA probes along the *E. coli* K-12 genome.

### 4.4. Gel Shift Assay

A gel shift assay was performed according to the standard procedure [36]. Probes for FlhDC-binding target sequences were generated via PCR amplification using a pair of primers (Appendix A) and Ex Taq DNA polymerase (TaKaRa). A mixture of each probe and FlhDC was incubated at 37 °C for 30 min in a gel shift buffer. After adding the DNA-loading solution, the mixture was subjected to PAGE. The gel was stained with GelRed (Biotium, Fremont, CA, USA) and detected using a LuminoGraph I (Atto, Amherst, MA, USA).

### 4.5. Consensus Sequence Analysis

An FlhDC-binding sequence set identified by the gSELEX-chip was analyzed using the MEME program [25] to analyze the FlhDC-binding sequence. Sequences were aligned, and a consensus sequence logo was created using WEBLOGO (http://weblogo.berkeley.edu/logo.cgi (accessed on 1 December 2022)).

### 4.6. Northern Blotting Analysis

Total RNA was extracted from the exponential phase of *E. coli* cells (OD_600_ = 0.4) in an M9 minimal medium supplemented with 0.2% glucose using an ISOGEN solution (Nippon Gene, Tokyo, Japan). RNA purity was verified by electrophoresis on a 1.5% agarose gel with formaldehyde, followed by staining with GelRed. Northern blotting analysis was performed as previously described [37]. DIG-labeled probes were prepared by PCR amplification using W3110 genomic DNA (50 ng) as a template with a pair of primers (Appendix A), DIG-11-dUTP (Roche, Basel, Switzerland), dNTP, gene-specific forward and reverse primers, and Ex Taq DNA polymerase. Total RNA (3 μg) was incubated in a formaldehyde-MOPS gel-loading buffer for 10 min at 65 °C for denaturation, subjected to electrophoresis on a formaldehyde-supplemented 1.5% agarose gel, and transferred onto a nylon membrane (Roche). Hybridization was performed using a DIG easy Hyb system (Roche) at 50 °C overnight with a DIG-labeled probe. The membrane was treated with anti-DIG-AP Fab fragments and CDP-Star (Roche) to detect the DIG-labeled probe, and the image was scanned using LuminoGraph I (Atto).

### 4.7. Measuring Glucose and Fructose Concentrations in the Culture Medium

*E. coli* cells were grown in an M9 minimal medium supplemented with 0.2% glucose or 0.2% fructose. A 0.5 mL aliquot of the bacterial culture was collected in a 1.5 mL tube and centrifuged at 20,400× *g* for 2 min. The supernatant was transferred to a new 1.5 mL tube for glucose and fructose measurements. The concentrations of glucose and fructose in the samples were measured using the ENZYTEC Liquid for D-Glucose/D-Fructose assay (R-Biopharm, Darmstadt, Germany).

## Figures and Tables

**Figure 1 ijms-24-03696-f001:**
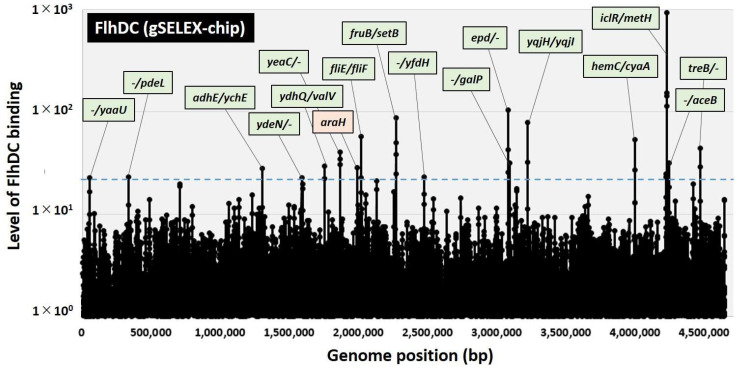
Identification of FlhDC-binding sites in the *E. coli* K-12 genome using gSELEX-chip. gSELEX screening of DNA-binding sequences was performed for FlhDC, a flagella master regulator TF of *E. coli*, using purified His-tagged FlhDC and a library of DNA segments from the *E. coli* K-12 W3110 genome. Following gSELEX, a collection of DNA fragments was subjected to gSELEX-chip analysis using the tiling array of the *E. coli* K-12 genome. The blue dotted line represents the cutoff level of twenty, and Table 1 lists the FlhDC-binding sites from cutoff level ten. Peaks shown in green represent the FlhDC-binding sites inside spacer regions, whereas peaks shown in orange represent the FlhDC-binding sites inside ORFs.

**Figure 2 ijms-24-03696-f002:**
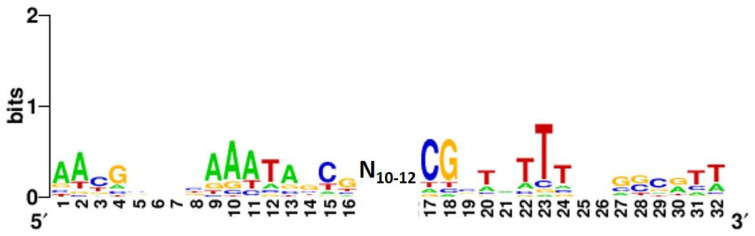
Consensus sequence of the FlhDC-box. Sequences of all the probes with high levels of FlhDC binding were analyzed using MEME (https://meme-suite.org/meme/ (accessed on 1 December 2022)) (see Table 1). WEBLOGO (http://weblogo.berkeley.edu/logo.cgi (accessed on 1 December 2022)) was used to perform matrix construction.

**Figure 3 ijms-24-03696-f003:**
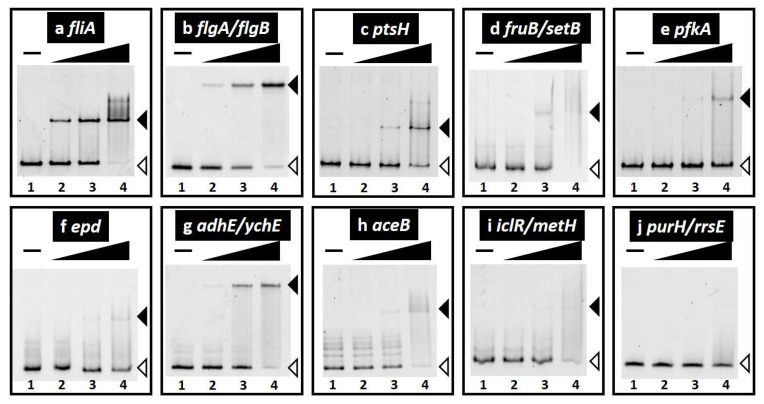
Gel shift assay of FlhDC–DNA complex formation. (**a**–**j**) Purified FlhDC was mixed with 0.1 pmol of each target DNA probe of the FlhDC-binding regions shown in Table 1 (also see Appendix A). FlhDC added: lane 1, 0 nM; lane 2, 10 nM; lane 3, 30 nM; lane 4, 100 nM. The filled triangles indicate the FlhDC–DNA probe complex, whereas open triangles indicate free probes.

**Figure 4 ijms-24-03696-f004:**
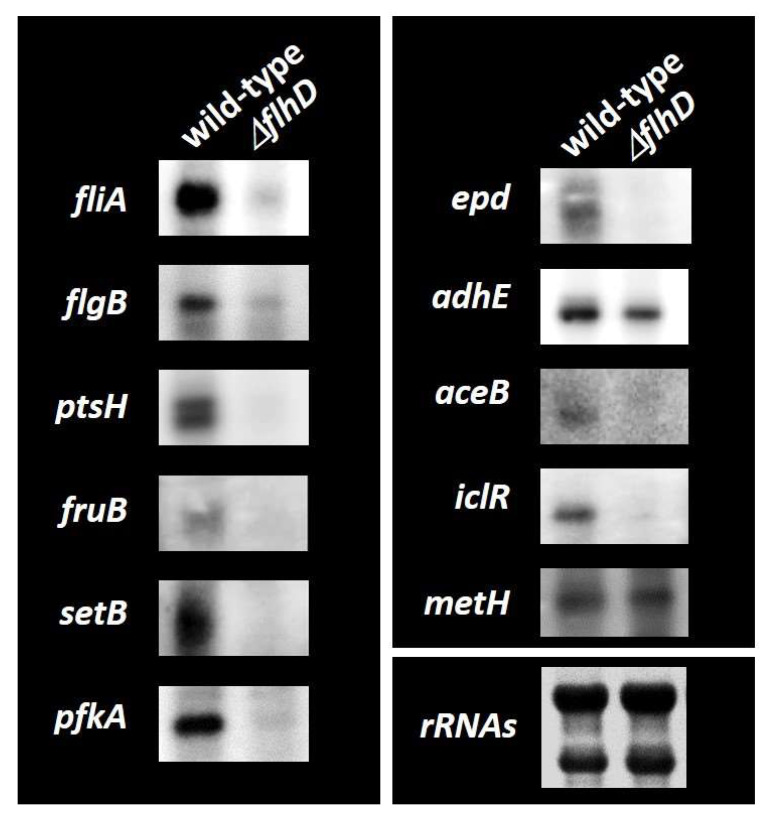
Northern blotting analysis of mRNAs from FlhDC target genes. Wild-type *E. coli* K-12 BW25113 and its *flhD* mutant JW1881 were grown in M9 0.2% glucose media. Total RNA was prepared during the exponential phase and subjected to Northern blotting analysis. DIG-labeled hybridization probes are shown on the left side of each panel. The amounts of total RNA analyzed were examined by measuring the intensity of ribosomal RNAs.

**Figure 5 ijms-24-03696-f005:**
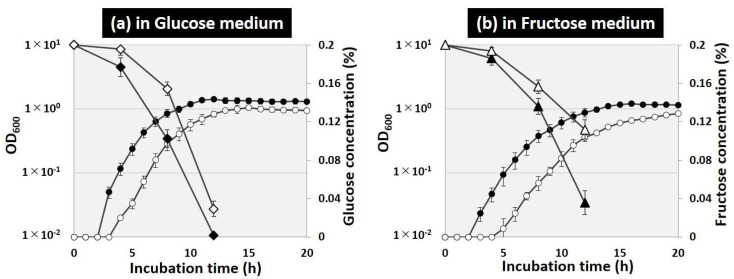
Effect of *flhD* deficiency on the growth curve and sugar consumption. *E. coli* BW25113 (closed symbols) and its *flhD* mutant JW1881 (open symbols) were grown in M9 media with 0.2% glucose (**a**) or 0.2% fructose (**b**). Cell density (circles in **a**,**b**) was monitored every hour and plotted along the time course. The concentrations of glucose (rhomboids in **a**) and fructose (rhomboids in **b**) were monitored every 4 h for up to 12 h.

**Table 1 ijms-24-03696-t001:** FlhDC-binding sites on the *E. coli* genome.

	Peak Position (bp)	Intensity	gSELEX Peak Type	Function	Operon	*gene*	D	FlhDC Site	D	*gene*	Operon	Function	FlhDC Box	position of FlhDC Box (bp)	Database (Known Target)
**[Inside spacer regions]**															
1	45772	21	B			*fixX*	>		>	*yaaU*	*yaaU*	predicted transporter	TACCCGTCAGCGGGCGTTTTCC ATCAGCTTTATTGCCGCGATG	45985.5	
2	331130	22	B			*betT*	>		>	*pdeL*	*pdeL*	predicted DNA-binding transcriptional regulator			
3	395546	10	A	beta-lactamase/D-alanine carboxypeptidase	*ampH*	*ampH*	<		>	*sbmA*	*sbmA-yaiW*	predicted transporter	TTCCGGGCAGATCCCGATTAG CGCCGCGCGTTTCTGGTCGTT	395756	
4	479234	13	B	maltose O-acetyltransferase	*maa*	*maa*	<		<	*hha*			AACGCATTAAATAATCGGTTTT CGTTAAAGGTTTTTCGGACATA	479262	
5	703072	18	A	glucosamine-6-phosphate deaminase	*nagBAC-umpH*	*nagB*	<		>	*nagE*	*nagE*	fused N-acetyl glucosamine specific PTS enzyme: IIC, IIB, and IIA components	AACGAGCCAAATAGGGTTCTC GTAGGGGGAATAAGATGAATA	702903	
6	791354	11	B	UDP-galactose-4-epimerase	*galETKM*	*galE*	<		<	*modF*			AAACAAACAACAATTGCGTT TCACCTTCGCTAATCAGCACATC	791409	
7	1057170	12	A	DNA-binding response regulator in two-component regulatory system with TorS	*torR*	*torR*	<		>	*torC*	*torCAD*	trimethylamine N-oxide (TMAO) reductase I, cytochrome c-type subunit	AACTCTGGAACGCGCTACGCCG ACCCAGTGCTCGTTGGTCGGTA	1057086	
8	1130058	13	A	assembly protein for flagellar basal-body periplasmic P ring	*flgAMN*	*flgA*	<		>	*flgB*	*flgBCDEFGHIJ*	flagellar component of cell-proximal portion of basal-body rod	AACGGCATAAATAGCGACCC ATTTTGCGTTTATTCCGCCGAT	1129912	*flgA, flgB*
9	1276852	11	A	sensory histidine kinase in two-component regulatory system with NarL	*narXL*	*narX*	<		>	*narK*	*narK*	nitrate/nitrite transporter	AACAACGCGGTCAACGTATTG CCAGCCGCAACCTGTGGATTT	1277030	
10	1297734	26	A	alcohol dehydrogenase	*adhE*	*adhE*	<		>	*ychE*	*ychE*	predicted inner membrane protein	AATGCTGTCAAAAGGCGTATT GTCAGCGCGTCTTTTCAACCTTA	1297508	
11	1488832	11	B			*aldA*	>		>	*cybB*	*cybB*	cytochrome b561	TAAGGCTGAAATACCCAACCC CGCCGATTATACCTAAGCCAAA	1488882	
12	1525236	11	B			*yncH*	>		>	*ydcD*	*rhsE-ydcD*	predicted protein			
13	1528470	11	B			*ydcD*	>		>	*ydcC*	*ydcC*	conserved protein	AAGATCATGAAAATTGTGATG TAAATCACGATTTTCATCTTT	1528271	
14	1570272	10	B	glutamate decarboxylase B, PLP-dependent	*gadBC*	*gadB*	<		<	*pqqL*			AACGGCAGTGTTAACATTCTCT ACCGTCATTTGTTTCAACAATT	1570260	
15	1580646	21	B	conserved protein	*ydeNM*	*ydeN*	<		<	*ydeO*			AAATAATCAAATAGCTAAAGC ATTCATCGTGTTGCCCGTATT	1580715	
16	1588868	18	B	predicted fimbrial-like adhesin protein	*ydeTSR*	*ydeS*	<		<	*hipA*			AATGTCGCGGATAAATTTTAT CGATTGCCGTTTTTTTGCCTTT	1588853	
17	1744232	28	A	conserved protein	*ydhQ*	*ydhQ*	<		>	*valV*	*valVW*	Val tRNA	AACGGTCGAAATAGCGCAGAA AATTACGTTTTGCCTCTTGCC	1744138	
18	1860030	38	B	conserved protein	*yeaC*	*yeaC*	<		<	*msrB*			ATCGTTTTTTCAACCGTTGATT TCTTCGCCGTTTTCGCCATC	1859802	
19	1986256	11	A	ncRNA	*isrB*	*isrB*	<		>	*yecR*	*yecR*	predicted protein	AATATTTTAATCAGCGAGGGGAT CTTCGCTGATTAAAGAAAT	1986340	*yecR*
20	1999848	10	B	RNA polymerase, sigma 28 (sigma F) factor	*fliAZY*	*fliA*	<		<	*fliC*			AATCGGACGATTAGTGGGTGAA ATGAGGGGTTATTTGGGGGTTA	1999646	*fliA*
21	2011168	54	A	flagellar basal-body component	*fliE*	*fliE*	<		>	*fliF*	*fliFGHIJK*	flagellar basal-body MS-ring and collar protein	AACGCCGTCCATAATCAGCCAC GAGGTGCGCGATGAATGCGACT	2010993	*fliE, fliF*
22	2017552	10	B			*fliK*	>		>	*fliL*	*fliLMNOPQR*	flagellar biosynthesis protein	AAAGCGCAGCAACGCGTCGTGC CCTCACCGGTCTTCTACGCGCT	2017546	*fliL*
23	2246656	16	B	lysine transporter	*lysP*	*lysP*	<		<	*yeiE*					
24	2261652	82	A	fructose-specific PTS enzymes: IIA component	*fruBKA*	*fruB*	<		>	*setB*	*setB*	lactose/glucose efflux system	GGCGTTTTTAATCGTTGCCTTT CTCACCGGTATTGCGGGCGCT	2261710	
25	2466256	22	B			*yfdG*	>		>	*yfdH*	*yfdGHI*	CPS-53 (KpLE1) prophage; bactoprenol glucosyl transferase			
26	2531672	13	B			*cysK*	>		>	*ptsH*	*ptsHI-crr*	phosphohistidinoprotein-hexose phosphotransferase component of PTS system (Hpr)	ATCTGGTTAAACTGATGGCGG AACTCGAGTAATTTCCCGGGTT	2531784	
27	2627634	10	B			*yfgG*	>		>	*yfgH*	*yfgHI*	predicted outer membrane lipoprotein	GGCGCGCTTATTGGCGCAGTC GCTGGCGGTGTTATCGGCCAC	2627801	
28	2859272	11	A	predicted DNA-binding transcriptional regulator	*ygbI*	*ygbI*	<		>	*ygbJ*	*ygbJK*	predicted dehydrogenase, with NAD(P)-binding Rossmann-fold domain	AATGCCTGCGCTACGTTGAAA GAGGCAGGTGCTTGCGGGGTT	2859340	*ygbJ*
29	2989270	11	A	predicted protein	*yqeK*	*yqeK*	<		>	*ygeG*	*ygeG*	predicted chaperone	GGGGCTAAAAATATCGATAAC GCAATGCAATGTTTCTATCACA	2989422	
30	3071934	98	B	D-erythrose 4-phosphate dehydrogenase	*epd-pgk-fbaA*	*epd*	<		<	*yggC*			CTTCGGCTACTTGCCGCGTTA ATCCTCCCGCAATTTTACGACTA	3071883	
31	3086262	30	B			*metK*	>		>	*galP*	*galP*	D-galactose transporter	TCGGGCGCAAAAAGAGCCTGAT GATCGGCGCAATTTTGTTTGTT	3086313	
32	3122256	12	B	conserved protein	*glcDEFGBA*	*glcG*	<		<	*glcF*			ATCGCCGCGAATTGCCAGTTTT TCCAGCGGTTCCTCGCGCAGA	3122357	
33	3132170	11	A	predicted protein with nucleoside triphosphate hydrolase domain	*yghSR*	*yghS*	<		>	*yghT*	*yghT*	predicted protein with nucleoside triphosphate hydrolase domain	CAGGCCGGAAATGTCGGTCGT GCAGTGACAAAATTACCGTTGAT	3132054	
34	3214830	74	A	predicted siderophore interacting protein	*yqjH*	*yqjH*	<		>	*yqjI*	*yqjI*	predicted transcriptional regulator	AACGCATCAAAGCGCGTTGCGT TGGCGCGGCGCTGCGCCAGAA	3215012	
35	3651858	14	A	IS5 transposase and trans-activator	*insH11*	*insH*	<		>	*slp*	*slp-dctR*	outer membrane lipoprotein			
36	3989170	50	A	hydroxymethylbilane synthase	*hemCDXY*	*hemC*	<		>	*cyaA*	*cyaA*	adenylate cyclase			
37	4213236	21	B			*metA*	>		>	*aceB*	*aceBAK*	malate synthase A	ACAGGCAACAACAACCGATGA ACTGGCTTTCACAAGGCCGTA	4213530	
38	4221834	873	A	DNA-binding transcriptional repressor	*iclR*	*iclR*	<		>	*metH*	*metH*	homocysteine-N5-methyltetrahydrofolate transmethylase	CACGCCAGAGAAACCGCGCTA CGTTGCCGGTGTTCTCGGCCCG	4221985	
39	4240352	10	B	D-xylose transporter	*xylE*	*xylE*	<		<	*malG*					
40	4281146	13	B	conserved protein	*yjcF*	*yjcF*	<		<	*actP*			ACTGCGGTAATCAGCCCCAGCC AGCCACCCATCATCGCGCCACG	4281280	
41	4414856	19	A	conserved protein	*bsmA*	*yjfO*	<		>	*yjfP*	*yjfP*	predicted hydrolase	GATGATTGAAATAGAATCACGC GAGCTGGCAGATATTCCCGTT	4414746	
42	4425648	11	B	predicted protein	*yjfZ*	*yjfZ*	<		<	*ytfB*					
43	4464250	42	B	trehalose(maltose)-specific PTS enzyme: IIB component/IIC component	*treBC*	*treB*	<		<	*treR*			CTTGCGCCAAGTGCCAGCGTGT CGGTTGCGCACAGTAAGGCGGT	4464299	
44	4638812	13	B			*yjjY*	>		>	*yjtD*	*yjtD*	predicted rRNA methyltransferase	GTCGGGCGAAATATCATTACT ACGCCACGCCAGTTGAACTGGT	4638978	
**[Inside genes, or downstrem of both transcription unit]**															
45	1092430	11	C	predicted outer membrane protein	*pgaABCD*	*pgaA*	<	*ycdT*	<	*insF*			GATATCTGGAATGGAGTGACG GTGATTGCTTTTTGTGCCGTA	1092524	
46	1223252	14	C			*ymgJ*	>		<	*minE*			AAGAATAGAAATATCGCCATC TTTTTGCTCAAGCTGTACGGTT	1223316	
47	1776868	10	C			*ydiO*	>	*ydiP*	>	*ydiQ*	*ydiQRST-fadK*	conserved protein	CCGGCGACAAACAGCGTTTCA CTGGCGTTATCAAAACAGCGTT	1777048	
48	1981072	27	C	trehalose-6-phosphate phosphatase, biosynthetic	*otsBA*	*otsB*	<	*araH*	<	*araG*					
49	2043166	14	C			*asnT*	>	*yeeJ*	>	*shiA*	*shiA*	shikimate transporter	ATCGGCCCAAAGCGTTGCCGA ACGTTTCGGTATTTCGGTGGCT	2042964	
50	2120630	20	C	predicted UDP-glucose lipid carrier transferase	*wcaCDEF-gmd-fcl-gmm-wcaI-cpsBG-wcaJ-wzxC*	*wcaJ*	<	*cpsG*	<	*cpsB*			AGCAGGTTAATTTTTTCATAT CGTTACCCTTTTTCAGGCAAT	2120756	
51	2732852	13	C	ncRNA	*ryfD*	*ryfD*	<	*yfiH*	<	*rluD*					
52	3134160	16	C			*yghT*	>	*pitB*	<	*gsp*					
53	3630156	11	C	predicted HlyD family secretion protein	*yhiI-rbbA-yhhJ*	*yhiI*	<	*yhiJ*	>	*yhiM*	*yhiM*	conserved inner membrane protein	AATATCGGTATTATCAAGAGA GCAACCGTAATTTTTGCTATT	3630007	
54	4236932	30	C			*yjbG*	>	*yjbH*	<	*yjbT*			AACCGGACATCTGACCGCCT ACTGGACGCCATCTTTCGCTCA	4237161	
55	4635274	13	C			*creB*	>	*creC*	>	*creD*	*creABCD*	inner membrane protein	TACTGACAACATTCTGACGCA AAATGCGCGTATGCAGGCATT	4635368	

## Data Availability

gSELEX data for FlhDC have been deposited in the ‘Transcription factor profiling of *Escherichia coli*’ (TEC) database at the National Institute of Genetics (https://shigen.nig.ac.jp/ecoli/tec/ (accessed on 13 December 2022)).

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
