# Peer review of "Genomic SELEX Reveals Pervasive Role of the Flagella Master Regulator FlhDC in Carbon Metabolism"

_ijms, 2023, doi:10.3390/ijms24043696_

Round 1

Reviewer 1 Report

My opinion is that manuscript shoud be published.

Author Response

Please see our response in the attached file.

Author Response

(The authors gave the same response as above.)

Reviewer 3 Report

The text is written in low-level English and in several places is insufficient to understand the meaning. There are a number of places in the manuscript that need improvement. It also appears that the text is chemically illiterate in some places. The text needs to be carefully checked and corrected. Below I have highlighted only some of these incomprehensible places with the following designations: <...> - for inclusion and ]...[ - for exclusion:

Introduction

44: The following abbreviation should be interpreted: TFs

Results:

180-181: The sentence is not comprehensible. It seems that some words have been left out. I have improved this sentence as follows. Is this correct?  "Total RNA  <was isolated>  from cells of wild-type E. coli K-12 and the flhD-deleted mutant   ]were[  grown in M9-glucose (0.2%) media."

207-209: Instead of a sentence, it makes no sense at all. The content is beyond comprehension. The authors need to rewrite this part: The cell density of the wild-type strain grown at OD600 reached 0.85after 8 h of incubation in a glucose sole carbon source medium and decreased accordingly to 0.1%, about half of the initial glucose concentration (Figure 5a).

214-216: The entire sentence is grammatically incorrect and incomprehensible: However, the effect of ..... than in the wild-type strain.

248-249: Again, the sentence is grammatically problematic: The new directly targeted FlhDC-regulated genes impact sugar utilization and carbon source metabolism.

Did the authors try to write the following: New genes regulated by FlhDC affect sugar utilization and metabolism of carbohydrate sources.

250: ]importation[   <import>

251: This is completely chemically incorrect: sugars, such as ATP and NADH,

272-274: The sentence is grammatically problematic: The glyoxylate pathway consists .... as a carbon substrate.

Author Response

(The authors gave the same response as above.)
